# Tubulin Carboxypeptidase Activity Promotes Focal Gelatin Degradation in Breast Tumor Cells and Induces Apoptosis in Breast Epithelial Cells That Is Overcome by Oncogenic Signaling

**DOI:** 10.3390/cancers14071707

**Published:** 2022-03-28

**Authors:** Trevor J. Mathias, Julia A. Ju, Rachel M. Lee, Keyata N. Thompson, Makenzy L. Mull, David A. Annis, Katarina T. Chang, Eleanor C. Ory, Megan B. Stemberger, Takashi Hotta, Ryoma Ohi, Michele I. Vitolo, Marie-Jo Moutin, Stuart S. Martin

**Affiliations:** 1Marlene and Stewart Greenebaum NCI Comprehensive Cancer Center, University of Maryland School of Medicine, 655 W. Baltimore St., Baltimore, MD 21201, USA; trevor.mathias@som.umaryland.edu (T.J.M.); jju@som.umaryland.edu (J.A.J.); leer@janelia.hhmi.org (R.M.L.); kethompson@som.umaryland.edu (K.N.T.); makenzy.mull@som.umaryland.edu (M.L.M.); david.annis@som.umaryland.edu (D.A.A.); ktchang@som.umaryland.edu (K.T.C.); eory@som.umaryland.edu (E.C.O.); megan.stemberger@som.umaryland.edu (M.B.S.); mvitolo@som.umaryland.edu (M.I.V.); 2Graduate Program in Molecular Medicine, University of Maryland School of Medicine, 800 W. Baltimore St., Baltimore, MD 21201, USA; 3Medical Scientist Training Program (MSTP), University of Maryland School of Medicine, 800 W. Baltimore St., Baltimore, MD 21201, USA; 4Department of Pharmacology and Physiology, University of Maryland School of Medicine, 655 W. Baltimore St., Baltimore, MD 21201, USA; 5Graduate Program in Epidemiology and Human Genetics, University of Maryland School of Medicine, 800 W. Baltimore St., Baltimore, MD 21201, USA; 6Graduate Program in Biochemistry & Molecular Biology, University of Maryland School of Medicine, 108 N. Greene St., Baltimore, MD 21201, USA; 7Department of Cell and Developmental Biology, University of Michigan, Ann Arbor, MI 48109, USA; hottat@umich.edu (T.H.); oryoma@umich.edu (R.O.); 8Grenoble Institut Neurosciences, University Grenoble Alpes, Inserm, U1216, CEA, CNRS, 38000 Grenoble, France; moutinm@univ-grenoble-alpes.fr; 9United States Department of Veterans Affairs, VA Maryland Health Care System, Baltimore, MD 21201, USA

**Keywords:** tubulin carboxypeptidase, detyrosinated tubulin, apoptosis, breast cancer, breast, cancer, invasion, tumor, microtubule

## Abstract

**Simple Summary:**

The recent discovery of the genetic identity of the tubulin carboxypeptidase (TCP) provides a unique opportunity to study the role of the detyrosination of α-tubulin (deTyr-Tub), as performed by the TCP, in breast epithelial cells and breast cancer cells. Previous research has shown that elevated deTyr-Tub conveys a poor prognosis in breast cancer and is upregulated in a coordinated manner at the invasive margin of patient tumor samples. Using TCP expression constructs, we have shown that increased deTyr-Tub promotes apoptosis in normal breast epithelial cells, that does not occur in the same cells with an oncogenic KRas mutation or Bcl-2/Bcl-xL overexpression. Furthermore, the addition of the TCP to the breast cancer cell lines MDA-MB-231 and Hs578t, also harboring Ras mutations, leads to increased focal gelatin degradation.

**Abstract:**

Post-translational modifications (PTMs) of the microtubule network impart differential functions across normal cell types and their cancerous counterparts. The removal of the C-terminal tyrosine of α-tubulin (deTyr-Tub) as performed by the tubulin carboxypeptidase (TCP) is of particular interest in breast epithelial and breast cancer cells. The recent discovery of the genetic identity of the TCP to be a vasohibin (*VASH1/2*) coupled with a small vasohibin-binding protein (*SVBP*) allows for the functional effect of this tubulin PTM to be directly tested for the first time. Our studies revealed the immortalized breast epithelial cell line MCF10A undergoes apoptosis following transfection with TCP constructs, but the addition of oncogenic KRas or Bcl-2/Bcl-xL overexpression prevents subsequent apoptotic induction in the MCF10A background. Functionally, an increase in deTyr-Tub via TCP transfection in MDA-MB-231 and Hs578t breast cancer cells leads to enhanced focal gelatin degradation. Given the elevated deTyr-Tub at invasive tumor fronts and the correlation with poor breast cancer survival, these new discoveries help clarify how the TCP synergizes with oncogene activation, increases focal gelatin degradation, and may correspond to increased tumor cell invasion. These connections could inform more specific microtubule-directed therapies to target deTyr-tubulin.

## 1. Introduction

The vast majority of cancer mortality is due to metastasis, the process by which cancerous cells spread to distant organs, and provides limited therapeutic options. In breast cancer specifically, greater than 95% of the mortality is associated with the spread of cancer to organs outside of the primary tissue [1]. The process by which cancerous cells migrate from the primary tumor and establish distant metastases is complex and multiphasic. In order to improve outcomes for cancer patients, it is important to better understand the underlying cellular processes involved in metastasis to develop new therapeutic interventions. Research into drugs that target the metastatic cascade is especially needed since nearly all current breast cancer treatments aim to reduce the size of established tumors by slowing cell growth or inducing cell death [2]. These therapeutics tend to take advantage of the fact that cancer cells divide more rapidly than many other tissues in the body. One common target of anti-neoplastic agents is the microtubule network to inhibit rapid proliferation by blocking mitosis [3]. Microtubule-targeted drugs are frequently used in cancer treatment and fall into two main categories: taxanes/epothilones, which promote microtubule stabilization, and vinca alkaloids/halichondrins, which promote microtubule depolymerization and collapse [3]. Microtubules are known to play an essential role in cellular processes such as replication, cell division, motility, cell polarity, and intracellular trafficking.

Microtubules are essential cellular structures formed by polymerizing stacks of α-tubulin/β-tubulin heterodimers that originate at the microtubule-organizing center (MTOC) in most mammalian cells. Microtubules are dynamic structures within the cell, and as they form and grow, post-translational modifications (PTMs) can be made to both α- and β-tubulin, modulating their stabilities and half-lives [4]. These modifications are known to play specific roles in animal cells, often with tissue- or cell type-specific functions [4]. Removal of the carboxy-terminal tyrosine of α-tubulin generates the PTM termed detyrosinated tubulin (deTyr-Tub), previously known as glu-tubulin. This tubulin PTM can then be further modified through the removal of revealed glutamic acid residues via a family of tubulin deglutamylases, CCP1-6, to generate Δ2 or Δ3 tubulin. Importantly, microtubules with increased stability often contain higher levels of deTyr-Tub, with those enriched in deTyr-Tub displaying an increased half-life of >1 h compared to 5–10 min for microtubules composed of the tyrosinated form of α-tubulin [5].

The detyrosination of α-tubulin has been known for more than 40 years [6], with two important enzymes composing the complete cycle of tyrosine removal and replacement, the tubulin tyrosine ligase (TTL) and the tubulin carboxypeptidase (TCP). The genetic identity of the TTL was first discovered in 1993 [7], but the genetic identity of the TCP remained elusive for decades. Research into the mechanism of action of the TTL revealed the TTL enzyme has affinity for α- and β-tubulin heterodimers, the building blocks of microtubules, rather than polymerized microtubules [8,9,10], whereas evidence has suggested that the TCP is acting on polymerized microtubules [11,12,13]. Despite not knowing the identity of the TCP, the resulting deTyr-Tub modification was able to be studied through direct observation of the PTM. Given the previously unknown identity of the TCP, very few compounds were available to specifically inhibit the formation of deTyr-Tub. One such compound is parthenolide, a sesquiterpene lactone isolated from the feverfew plant. It has been previously shown that parthenolide decreases levels of deTyr-Tub; however, determining a specific mechanism by which this drug modulates tubulin is hampered by its broad-spectrum anti-inflammatory effect [14]. One study revealed the ability of parthenolide to bind with *VASH1* via crystal structure [15], while another has suggested that a specific enantiomer of parthenolide is responsible for its function in reducing deTyr-Tub [16]. Recent work determined that parthenolide binds covalently to microtubules to non-specifically prevent the generation of deTyr-Tub [17].

When examining the role of deTyr-Tub in cancer development and progression, research performed prior to the discovery of the identity of the TCP revealed a connection between detyrosinated microtubules, poor patient prognosis, and connection to migratory phenotypes. Early research revealed that microtubules composed of deTyr-Tub orient towards a wound edge in vitro using a scratch assay [18]. Before the discovery of the TCP, levels of deTyr-Tub were studied in primary breast tumors. One such study examined the relationships between elevated deTyr-Tub and other clinical markers in patients with all types of breast cancer (ER+, PR+ and TNBC) revealing a correlation with a high pathologic mitotic score [19]. In this somewhat limited dataset of breast cancers, patient outcomes were compared between those with high and low expression of deTyr-Tub and (*n* = 134), 46 patients had adverse outcomes following initial treatment and elevated deTyr-Tub trended with worse prognosis, but the difference was not statistically significant (*p* = 0.27). When deTyr-Tub status was combined with tumor grade (high deTyr-Tub score and high grade), patients were statistically more likely to have a poor prognosis (*p* = 0.02). These findings paved the way for additional studies in the breast cancer field. More recently, expression of deTyr-Tub has been examined in circulating tumor cells (CTCs) isolated from patients with breast cancer [20,21]. In one study, levels of deTyr-Tub, along with other cytoskeletal markers, were compared in CTCs isolated from patients with early-stage breast cancer to those from patients with metastatic disease [21]. This study identified that CTCs from patients with metastatic disease expressed higher levels of deTyr-Tub, total α-tubulin, and vimentin compared to those with early-stage disease. Similar patterns of cytoskeletal markers have been seen in breast cancer cells which have undergone epithelial-to-mesenchymal transition (EMT) [22].

The complex mechanisms of metastatic dissemination are not fully known; however, the mechanism of EMT is thought to play an important role [23]. EMT is a biologically conserved process which occurs during embryonic development, where cells lose their tight cell-to-cell contacts in order to migrate in the developing organism [24]. Research has shown that when breast cancer cells are forced to undergo EMT via exogenous expression of the mesenchymal transcription factors, Twist or Snail, levels of deTyr-Tub increase. Conversely, this effect can be observed through the suppression of Twist in mesenchymal-like cells, leading to a decrease in the levels of deTyr-Tub. Increased levels of deTyr-Tub are observed in cells undergoing EMT at the invasive fronts of patient breast tumors, where invasion of adjacent tissues is starting to occur [25]. These findings complement much earlier work which revealed cells at a wound edge generate polarized deTyr-Tub-rich microtubules oriented in the direction of the wound [18,26].

All of these earlier studies into the function and regulation of deTyr-Tub were hampered by the inability to target the unknown TCP enzyme and this has limited our understanding of the role of TCP in cancer. However, the major TCP enzymes were recently discovered to be vasohibin 1 (*VASH1*) or vasohibin 2 (*VASH2*) coupled with a small vasohibin-binding protein (*SVBP*) in late 2017 [27,28]. Recent publications have shown the importance of SVBP as a core element essential for tubulin detyrosination [29,30]. The discovery of the TCP to be *VASH1* or *VASH2* coupled with *SVBP* has revealed the mechanism by which α-tubulin is converted to deTyr-Tub. This detyrosinated α-tubulin can subsequently have the tyrosine replaced by the TTL. Given our current understand of the tubulin detyrosination cycle and the enzymes involved allows for the functional effects of this tubulin PTM to be directly tested for the first time.

The process by which deTyr-Tub and EMT are connected remains unknown; however, recent studies show the inhibition of *VASH2* expression through siRNA leads to a restoration of E-cadherin [31]. This work was performed in breast cancer cell lines that are predominantly mesenchymal in phenotype and this manipulation forces these cells to undergo a mesenchymal-to-epithelial conversion. This observation, combined with the recent elucidation of the role of VASH/SVBP as the TCP, implies that microtubule modifications may play an essential role in the EMT process.

Existing knowledge of the role of deTyr-Tub in cells coupled with the recently established genetic identity of the TCP allows for us to directly increase deTyr-Tub through the systematic addition of the TCP components. Transfection of TCP constructs into immortalized breast epithelial cells induced apoptosis that was suppressed by activated *KRas*, Bcl-2 and Bcl-xL overexpression. Breast cancer cells could tolerate elevated TCP expression and demonstrated greater tumor cell invasion. Together, these results demonstrate that oncogenic signaling pathways can enable tolerance of an elevated level of deTyr-Tub and that TCP transfection promotes increased invasiveness.

## 2. Materials and Methods

### 2.1. Antibodies and Reagents

Bcl-xL antibody (1:1000, Cat#: ab32370) and VASH1 (1:1000, Cat#: ab199732) were purchased from Abcam. Bcl-2 antibody (1:1000, Cat #: 15071S), and Cleaved Caspase-3 (1:1000, Cat#: 9661S) were purchased from Cell Signaling Technologies. α-tubulin (1:1000, Cat#: T6199), FLAG (1:1000, Cat#: F3165), Tyr-Tub antibody (1:1000, Cat# MAB1864-I), and VASH2 (1:1000, Cat#: MABC536) were purchased from Sigma. GAPDH (1:5000, Cat#: sc-32233) was purchased from Santa Cruz. SVBP (CCDC23, 1:1000, Cat#: PA5-52569) was purchased from Invitrogen. The deTyr-Tub antibody was developed by Takashi Hotta and Ryoma Ohi [17]. This antibody is now commercially available through RevMab under catalogue number RM444 (1:10000). Alexa Fluor 568 (1:1000, Cat#: A11011), Alexa Fluor 594 (1:1000, Cat#: A11012) and Hoechst 33258 (1:5000, Cat#: H3569) were purchased from Invitrogen (Waltham, MA, USA).

### 2.2. Cell Culture

MCF-10A cells were purchased from ATCC and cultured in DMEM/F-12 Media (Invitrogen, Cat#10565-018) supplemented with 5% Horse Serum (Invitrogen, Cat# 26050-088), 1% Pen/Strep (Gemini Bio-products, Cat# 400-109), Recombinant Human EGF (Invitrogen, 100 µg/500 mL), Hydrocortisone (Sigma H-0135, 50 µg/mL), Cholera Toxin (Sigma C-8052, 50 µg/500 mL), and Insulin (Sigma I-9278, 10 µg/mL). The parental MCF-10A cells were not passed beyond 35 passages in our laboratory, in accordance with protocols published from the Brugge Laboratory, to preserve their growth profile, and sensitivity to apoptosis and anoikis [32].

MCF7 and MDA-MB-231 cells were purchased from ATCC and cultured in DMEM Media supplemented with 10% FBS (Atlanta Biologicals, R&D Systems, Minneapolis, MI, USA) and 1% Pen/Strep (Gemini Bio-products, Cat# 400-109).

MCF10A cells with homozygous loss of *PTEN* or stable expression of activated KRas have been described previously, and characterized for effects on primary tumor formation and metastasis. The MCF10As with activated KRas pathway are able to form tumor in vivo and show a strong upregulation of p-ERK [33,34,35]. MCF10A cells with stable expression of Bcl-2 and Bcl-xL were generated using lentiviral transduction of Bcl-2 or Bcl-xL plasmids containing a puromycin selection vector obtained from addgene (see ‘Lentivirus production and transduction’ below).

### 2.3. Lentivirus Production and Transduction

Overexpression of Bcl-2 and Bcl-xL in MCF10A cells was performed by lentiviral transduction using the Lenti-X Packaging System (Clontech, Mountain View, CA, USA) according to the manufacturer’s instructions. The pCDH-puro-Bcl2 plasmid (Addgene plasmid #46971) and pCDH-puro-Bcl-xL (Addgene plasmid #46972) were added into supplied nanoparticle complexes for 10 min and applied to Lenti-X 293T cells to produce the virus. The medium was changed after 24 h and viral supernatant was harvested after 48 h, filtered, and used to infect cells at an approximate MOI of 10 along with 1 µg/mL Polybrene. The plate was then immediately centrifuged for 1.5 h at 1000 rpm. Cells were selected with 1 µg/mL puromycin at 2 days after infections and maintained in puromycin-containing media. Puromycin was removed from the medium for experimental conditions. pCDH-puro-Bcl2 and pCDH-puro-Bcl-xL were gifts from Jialiang Wang [36].

### 2.4. Transient Transfections

#### TCP and Individual Component Plasmids

FLAG-sfGFP-His

FLAG-*VASH1*-sfGFP-His

FLAG-*VASH2*-sfGFP-His

FLAG-*SVBP*-myc

FLAG-*VASH1*-sfGFP-His + IRES + *SVBP*-myc

FLAG-*VASH2*-sfGFP-His + IRES + *SVBP*-myc

These plasmids for expression of the TCP components were generously provided through our collaboration with the Moutin group [27].

Transfections were performed 24 h after plating cells at the indicated density for each given experiment after being harvested from a ~80–90% confluent tissue culture dish. The plasmid of interest was prepared at a concentration of 2 µgDNA/100 µL at a ratio of 1 µg to 3 µL of FuGene HD transfection reagent (Promega, Cat#: E2311).

### 2.5. Live Imaging

Epi-fluorescence images of transfected cells were captured with a Nikon Ti2E microscope with High Content Analysis (HCA). Images were collected with equal exposure times between conditions. A 10× phase-contrast objective was utilized to capture images of the central 5 mm square of each well on the IBIDI 8-well chamber slide with IBITreat (Fitchburg, WI, USA; Cat.No:80826) and images were stitched together by the Nikon Elements software AR 5.21.03 (Nikon). Using the GFP channel of these images, GFP+ objects were detected using the Nikon GA3 analysis suite. Objects were determined based on having an intensity greater than 12,000 RFUs and a size greater than 10 µm but less than 150 µm to identify individual cells rather than cellular debris or aggregates, respectively. A 2× clean and separate filter was employed to remove additional debris and separate adjacent cells into unique objects. With individual GFP+ objects identified, the number of objects was enumerated using the Nikon General Analysis software, as well as the cell length and circularity for each object.

### 2.6. Single Cell Imaging and Tracking

MDA-MB-231 or Hs578t cells were transfected with sfGFP or TCP constructs and moved to IBIDI 8-well chamber slide with IBITreat (Fitchburg, WI, USA; Cat.No:80826) 12 h after transfection and allowed to attach for 12 h before imaging. A total of 1 to 2 fields of cells were imaged in each condition using a 10× phase-contrast objective in GFP and brightfield channels every 15 min for 19 h. Images were processed using the ImageJ plugin TrackMate [37] to identify GFP+ cells and track them across the time series. Tracks were filtered based on a minimum duration of 5 h with no maximum duration in length. These tracks were exported and analyzed using the MotilityLab platform [38]. Plots were generated for mean squared displacement (MSD) and mean track speed in MotilityLab using the aggregate data across biological and technical replicates.

### 2.7. Immunofluorescence

Cells were fixed using freshly prepared 3.7% formaldehyde diluted in 1x phosphate-buffered saline (PBS) for 10 min. The fixation reagent was removed and 2 washes of 1× PBS were performed 5 min apart. Fixed cells were permeabilized using a 0.1% *v*/*v* ratio of Triton-X100 in PBS. The permeabilization solution was left on the cells for 10 min followed by 2 washes of PBS 5 min apart. Cells were blocked in a solution containing 5% bovine serum albumin (BSA) *w*/*v*, a 0.5% *v*/*v* ratio of NP-40 substitute in PBS for one hour at room temperature while being protected from light. Primary antibodies were added at the outlined ratios in primary antibody solution consisting of a 2.5% BSA *w*/*v* ratio, with 0.5% *v*/*v* ratio of NP-40 substitute in 1× PBS. Primary antibodies were allowed to bind overnight at 4 degrees Celsius with gentle rocking. Removal of unbound and non-specific antibody was accomplished with 3 consecutive washes with PBS 10 min apart with gentle rocking during the incubation steps. Secondary antibodies conjugated to indicated Alexa fluorophores were diluted in fresh primary antibody solution and diluted at 1:1000, for nuclear co-staining a 1:5000 dilution of Hoescht 33258 is added to the secondary antibody solution. Cells were stained for 2 h at room temperature. Removal of unbound and non-specific antibodies was accomplished with 3 consecutive washes with PBS 10 min apart with gentle rocking during the incubation steps. A final quick wash with double-distilled water was performed before adding Fluoromount-G (Invitrogen, Cat# 00-4958-02, Waltham, MA, USA).

### 2.8. Active Caspase-3 Immunofluorescence Assay

Cells were seeded at 15,000 cell/well onto IbiTreat 8-well chambers and allowed to adhere for 24 h. They were subsequently transfected with sfGFP, TCP-*VASH1*, or TCP-*VASH2* plasmids for 24 h. The cells were then fixed, permeabilized and stained as described above with a Cleaved Caspase-3 antibody and an Alexa Fluor 594 secondary antibody. Images were acquired on a Nikon Ti2-E inverted microscope at 10× magnification. To analyze these images, first, the boundary of a successfully transfected cell was identified using the GFP channel in Nikon HCA to create an object for each cell. The same metrics to identify GFP+ objects were utilized as from the imaging of transfected live cells. With each object identified, the boundary of the object was slightly eroded to ensure that the AF549 intensity associated with Caspase-3 is only a result of that cell, and not any adjacent cells or debris. The average AF594 intensity of the cell is then calculated for each eroded object. Cells were determined to be positive for active Caspase-3 when the average 594 intensity of the GFP+ object was calculated to be >10,000 RFUs.

### 2.9. Western Blotting

Cells were lysed at the indicated time point using a chilled solution of 1× RIPA containing phenylmethylsulfonyl fluoride, protease inhibitor cocktail, and phosphatase inhibitor cocktail II. Dishes or plates containing the cells of interest were placed on a bed of ice for 15 min following the addition of the RIPA solution. Cells were scraped from the dish using a cell scraper and lysates collected in a 1.5 mL microcentrifuge tubes. These cells were vortexed every 5 min for the next 15 min, and then placed in the −30 °C freezer until ice crystals formed. The tubes were then thawed on wet ice and centrifuged at 15,000 rcf for 15 min. The supernatants were removed and placed in a new microcentrifuge tube, while the resulting pellets were discarded. The resulting total protein was quantified using the BioRad DC Protein Assay Kit according to the manufacturer’s recommended protocol. Samples and standards were read on a BioTek Synergy HT plate reader after a 15 min incubation period. Samples were diluted to a final concentration of 1 µg/µL, boiled at 95 °C for 10 min, and loaded in gels at equal volumes and total protein quantity.

An amount of 20 µg of total protein was added to each lane of a 1.5 mm × 10 well NuPAGE 4–12% Bis-Tris Gel (Invitrogen, Cat# NP0335BPX). Loaded gels were run using 1× NuPAGE MES SDS Running Buffer using 2 constant voltage phases, the first phase is 90 volts for 30 min, followed by 120 volts for 90 min. Gels are trimmed and transferred to a PVDF membrane using the eBlot™ L1 Fast Wet Transfer System (GenScript, Piscataway, NJ, USA). Membranes were blocked in either 5% BSA in 1× TBST or 5% non-fat dry milk in 1× TBST rocking for 1 h at room temperature. Primary antibodies were added to 2.5% BSA or non-fat dry milk in 1× TBST shaking overnight at 4 °C. Removal of unbound and non-specific antibody was accomplished with 3 consecutive washes of 1× TBST shaking for 10 min each. HRP-conjugated secondary antibody was diluted at 1:5000 in 2.5% BSA or non-fat dry milk in 1× TBST and incubated for 2 h at room temperature with gentle rocking. Removal of unbound and non-specific antibody was accomplished with 3 consecutive washes of 1× TBST rocking for 10 min each. Electrochemiluminescence reagent (Amersham) was added to blots for 2 min before capturing images on the iBright imager (ThermoFisher, Waltham, MA, USA).

### 2.10. Chemiluminescence and Molecular Weights

Using the iBright Software (ThermoFisher, Waltham, MA, USA), the molecular weights of bands of interest can be determined. For each blot, the SeeBlue™ Plus2 Pre-stained Protein Standard (LC5925, Invitrogen, Waltham, MA, USA) was used. The software allows for each band of the marker to be identified. Using these as the standard, the molecular weight of a band of interest is calculated by the software.

### 2.11. Gelatin Degradation Assay

The Cy-3-labeled gelatin degradation assay kit (Sigma, Cat#: ECM671) was purchased from EMD Millipore and the manufacturer’s protocol was followed to generate gelatin coated 8 chamber IBIDI well slides (Cat: 80826, poly-L-lysine coated). To generate a more consistent layer of fluorescent gelatin, 100 µL of prepared gelatin was added to each well and constantly rotated during the 5 min solidification time in the protocol. MDA-MB-231 cells were transfected 24 h prior to plating on prepared gelatin and cells were fixed using 3.7% formaldehyde diluted in 1× PBS after 24 h on the gelation. Hs578t cells were transfected 24 h prior to plating on prepared gelatin, and cells were fixed 48 h after plating using 3.7% formaldehyde diluted in 1× PBS. A total of 10 GFP+ cells were imaged at random on an Olympus FV1000 point-scanning confocal microscope at 60× magnification from each well to generate 20 or 30 images per condition per biological replicate. Four biological replicates were performed. A total of 110 cells were imaged per condition per cell line.

For each image, degradation percentages were calculated in MATLAB based on a maximum intensity projection (MIP) of the original images. GFP MIP images were filtered with multiple sizes of Laplacian of Gaussian filters ranging from a sigma of 2 pixels to 20 pixels, increasing in 1 pixel increments. A maximum intensity projection across the filtered images resulted in an image highlighting the cell body; a threshold for this image was set at values greater than the 75th percentile of the filtered maximum projection. Objects less than 10 pixels were removed and values in a 20 pixel border around the edge of the image were set to 0 to remove edge effects. This image was further process by closing the binary image with a disk of radius 4 pixels, filling holes in the image, and then removing objects less than 50,000 pixels. The MIP of the Cy-3 images were contrast adjusted using the MATLAB function imadjust (default of 1% saturation) and then the image was binarized to keep values less than 10,000. A further constraint that the original intensity be less than 1200 was then added and objects smaller than 10 pixels were removed. Percent degradation was calculated as the number of degraded pixels in the Cy-3 image that were inside the cell boundary, divided by the number of pixels inside the cell boundary.

After cells were fixed, one slide per cell line was permeabilized for 10 min with 0.25% TritonX100 in PBS. These cells were then stained with phalloidin conjugated to AF647 at a 5× stock concentration in PBS for 30 min. Cells were washed 3 times with PBS and preserved using Fluoromount-G. Cells were imaged on an Olympus FV1000 point-scanning confocal microscope at 60× magnification and representative overlay of image channels were generated using ImageJ.

### 2.12. Statistical Analysis

GraphPad Prism (version 9) was used to determine all statistical comparisons. One-way and two-way ANOVA tests were performed with a Tukey multiple comparisons post-test as indicated. A *p*-value of 0.05 or less was considered statistically significant. Multiple technical replicates were averaged for each experiment, and the means were compared across biological replicates.

## 3. Results

### 3.1. VASH1/2 in Combination with SVBP Increase Tubulin Detyrosination in Breast Epithelial Cells and Breast Cancer Cell Lines

The plasmids developed by the Moutin group in their manuscript that reported the discovery of TCP were used for transfection experiments [27]. The plasmid constructs under CAG promoters are as follows: FLAG-sfGFP, FLAG-*VASH1*-sfGFP, FLAG-*VASH2*-sfGFP, and FLAG-*SVBP*-myc-tag as well as FLAG-*VASH1*-sfGFP and FLAG-*VASH2*-sfGFP plasmids with an internal ribosomal entry site (IRES) for *SVBP*-myc-tag. We confirmed that the TCP constructs, TCP-*VASH1* or TCP-*VASH2* that express both the enzymatically active vasohibin and *SVBP* under an IRES, specifically increase deTyr-Tub in breast epithelial cells (Figure 1). The combination constructs containing *VASH1* or *VASH2* plus IRES *SVBP* will be referred to as TCP-*VASH1* or TCP-*VASH2*. Transfection of these plasmids leads to a specific increase in filamentous deTyr-Tub in those cells expressing the *VASH-sfGFP* fusion protein in MCF10A (Figure 1A). Immunofluorescence demonstrates that elevated deTyr-Tub appears in the polymerized, filamentous microtubules within transfected MCF10A cells. Immunoblotting confirms an increase in deTyr-Tub in MCF10As at 16 h post-transfection, reaching a maximum at 24 h post-transfection, which begins to dissipate at 48 h post-transfection (Figure 1B). The reduction from peak levels of deTyr-Tub at 48 h is more profound in the *VASH2* TCP-transfected cells than the *VASH1* TCP-transfected cells for breast epithelial cells. The *sfGFP*, *VASH1*-*sfGFP*, *VASH2*-*sfGFP*, all contain an N-terminal FLAG-tag, probing with a FLAG-antibody allows for the detection of all three proteins at their respective molecular weights. A decrease in VASH1 and VASH2 can be seen at a similar time as the decrease in deTyr-Tub, but the loss of the transfected protein does not occur with FLAG-tagged sfGFP.

Similarly, the breast cancer cell line MCF7 displays an increase in filamentous deTyr-Tub (Figure 1C). The time course following transfection of MCF7s shows an increase in deTyr-Tub beginning at 16 h and continues through the 48 h time point for both TCP constructs. The amount of FLAG-tagged *VASH1* and *VASH2* protein expression increases less over time compared to the increase in *sfGFP*. These data reveal that transfection of the TCP constructs can increase deTyr-Tub in breast epithelial cells and breast cancer cells, but in epithelial cells that increase is transient and protein levels drop rapidly compared to sfGFP control (Figure 1D). The conversion of Tyr-tub to deTyr-Tub in cells transfected with the TCP constructs can be visualized by immunostaining performed in Appendix A. The same cell lysates from Figure 1 were probed for Tyr-Tub in Appendix A.

### 3.2. Expression of Both TCP Components Robustly Increases deTyr-Tub in MCF10As and MCF7s

To assess the effect of the individual TCP components or the combination of components, MCF10As and MCF7s were transfected with sfGFP control, TCP-*VASH1*, TCP-*VASH2*, *VASH1* alone, *VASH2* alone, or *SVBP* alone and lysates collected at 24 h post-transfection. The greatest increase in deTyr-Tub can be observed in the TCP-*VASH1* and TCP-*VASH2* constructs, while the constructs with *VASH1* and *VASH2* alone moderately increase deTyr-Tub. This effect can be seen in both the breast epithelial cell line, MCF10A (Figure 2A,B), and breast cancer cell line, MCF7 (Figure 2C,D). Furthermore, constructs expressing *sfGFP* and *SVBP* are comparable in their basal level of deTyr-Tub (Figure 2B,D). The slight difference in molecular weight for the SVBP expressed under the IRES compared to that expressed alone is due to a His-tag present in the *SVBP* alone construct. Despite similar levels of vasohibin as observed by FLAG immunoblotting, the TCP constructs which co-express *SVBP* are the most effective at increasing deTyr-Tub. When examining the effect of the TCP components via immunofluorescence, the combination of vasohibin and *SVBP* detyrosinates the vast majority of filamentous microtubules within the cell (Figure 1A and Appendix A). However, vasohibin alone only detyrosinates a subset of microtubules, and *SVBP* alone has no effect on the abundance of deTyr-Tub (Figure 2B,D).

### 3.3. TCP Expression Leads to Changes in Cell Morphology Suggestive of Apoptosis

When imaging MCF10A cells transfected with the TCP constructs, several cells displayed visual characteristics of apoptosis, apoptotic vesicles staining for deTyr-Tub and the sfGFP-tagged vasohibin, as well as nuclear consolidation in a subset of cells (Figure 3A). The decrease in transfected vasohibin protein levels compared to *sfGFP* (Figure 1B), and visualization of apoptotic morphologies were indicative of possible programmed cell death. To test this, we transfected MCF10A cells and imaged them in both phase contrast and 488 nm channel at 16, 24, and 48 h after transfection using a Nikon Ti2E with High Content Analysis (HCA) (Figure 3B). The number of transfected cells, size, and circularity were determined for *sfGFP*-positive objects at 24 h and 48 h post-transfection. Interestingly, MCF10A cells expressing the TCP-*VASH1* and TCP-*VASH2* were significantly less numerous, smaller in length, and more circular (Figure 3C), when compared to control cells expressing sfGFP. Using a minimally aggressive breast cancer cell line, MCF7, transfected with the same plasmid constructs resulted in a similar trend and cellular morphology (Figure 3D). Like in the breast epithelial cells, MCF7 cells transfected with the TCP constructs become smaller and less numerous compared to the *sfGFP* control vector. Interestingly, no significant change in cell circularity was observed in the MCF7s, likely due to the cells having a more circular morphology at baseline.

### 3.4. TCP Transfection-Induced Apoptosis Is Prevented through Oncogenic Signaling

These data above suggested that the TCP constructs may be inducing apoptotic cell death in transfected cells, so we developed a method utilizing immunofluorescence to stain transfected cells with an active Caspase-3 primary antibody (CST-9661S) and an AF594-conjugated secondary antibody. Transfected cells were fixed, permeabilized, and stained for the presence of active Caspase-3. These cells were imaged at 488 nm and 594 nm for the vasohibin–*sfGFP* fusion protein and active Caspase-3, respectively. The distribution of AF594 intensities for *sfGFP*+ objects is shown in Appendix A, with the baseline cut off shown as a red vertical line. This same cut off was used for all MCF10A variant cell lines to determine the apoptotic double-positive cells.

Elevated deTyr-Tub in patient tumors and breast tumor cell lines indicates that cancerous cells can tolerate increased deTyr-Tub [17,30], so we tested the role of specific anti-apoptotic and oncogenic signaling mutations in the stable genetic background of MCF10A cells [26,27,28]. These MCF10A cells with specific oncogenic alterations were then transfected with control *sfGFP* vector, TCP-*VASH1*, or TCP-*VASH2* constructs for 24 h prior to fixation and staining with the active Caspase-3 antibody. Using the same analysis method and cut offs as with the MCF10A parental cell line, percentages of active Caspase-3-positive cells were quantified for each cell line. There was an increase in Caspase-3 positivity in the each MCF10A variant expressing both TCP-*VASH1* and TCP-*VASH2* except for the cell lines expressing the oncogenic KRas. Somewhat unexpectedly, the introduction of oncogenic *KRas* into the MCF10As cell line prevented the induction of apoptosis following TCP transfection as measured by Caspase-3 activation (Figure 4A). To determine whether the oncogenic variations and/or anti-apoptotic proteins can prevent Caspase-3 activation induced by TCP-*VASH1* or TCP-*VASH2*, the percentage Caspase-3 positivity per cell line was graphed (Figure 4B). The loss of *PTEN* has been shown to prevent apoptosis caused by numerous stimuli, such as growth factor deprivation, anchorage independence, and disruption of the actin-cortex via latrunculin-A treatment [28]; however, there was no significant difference in apoptosis between the parental MCF10A cells and the *PTEN*−/− cells transfected with TCP-*VASH1* or TCP-*VASH2* (Figure 4B). In this assay, TCP-*VASH1* expression in cells expressing oncogenic *KRas* (alone or in combination with *PTEN* loss) and overexpressing Bcl-xL showed a significant decrease in the amount of Caspase-3 positivity compared to the MCF10A cells. TCP-*VASH2* expression in these cell lines showed cells expressing the oncogenic *KRas* (alone or in combination with *PTEN* loss), Bcl-xL and Bcl-2 overexpression resulted in a significant decrease in the amount of Caspase-3 activation compared to the MCF10A cells.

### 3.5. TCP Components Do Not Correlate with deTyr-Tub in MCF10A Variants and Breast Cancer Cells

In an effort to understand the underlying connection between the introduced oncogenic and anti-apoptotic signals in MCF10As and the mechanism of alpha tubulin detyrosination, we probed cell lysates for deTyr-Tub, the components of the TCP (*VASH1*, *VASH2*, and *SVBP*), as well as total alpha-tubulin as a control. Interestingly, despite an increase in deTyr-tubulin in the *KRas* and *PTEN*−/− *KRas* cells, as well as the Bcl-2/Bcl-xL-overexpressing line, there was minimal change in the protein levels of the components of the TCP (Figure 5A). We wanted to determine the abundance of deTyr-Tub and TCP components in commonly utilized breast cancer cell lines. We observed that the breast cancer cell lines, T47D, BT-549, and MDA-MB-436 displayed high levels of the deTyr-Tub PTM. Despite the elevation in deTyr-Tub, the TCP components in these cell lines were not different from the other cell lines with low levels of the tubulin PTM (Figure 5B). These data suggest that the regulation of deTyr-Tub may be mediated through a mechanism independent of protein expression of the TCP enzyme components or through a different, yet to be discovered tubulin carboxypeptidase.

### 3.6. Increased deTyr-Tub Enhances Gelatin Degradation

Given our prior work showing the elevation of deTyr-Tub at the invasive margin of patient breast cancer tissue samples [17], we aimed to understand if enhanced deTyr-Tub promotes a more invasive phenotype. To test tumor cell invasiveness, transfected MDA-MB-231 and Hs578t breast cancer cells were plated on Cy-3-labeled gelatin and the amount of gelatin degraded under the cell body was analyzed. To examine whether TCP-*VASH1* or TCP-*VASH2* could promote tumor cell invasion, the tumorigenic MDA-MB-231 and Hs578t breast cancer cells were used. It is known the MDA-MB-231 and Hs578t cells have all of the necessary machinery to degrade gelatin, whereas the MCF10A and variant cell lines were not able to perform this task at baseline. Given the inability of the MCF10A cell line to perform this task at baseline, we selected MDA-MB-231s and Hs578ts to proceed with the invadopodia assay. Cells were transfected with TCP-*VASH1* and TCP-*VASH2* or *sfGFP* control. Areas of degradation are seen by the dark areas on the representative Cy3 gelatin images, denoting the absence of fluorescent gelatin (Figure 6A,C). MDA-MB-231 cells effectively degraded the gelatin matrix in a punctate pattern as observed in the representative images in Figure 6A. The Hs578ts were also able to degrade the gelatin, but to a lesser extent than the MDA-MB-231s. Given this slower degradation rate, the Hs578ts were not fixed until 48 h after plating, rather than 24 h. The percentage of degradation for each cell (as defined by degraded gelatin area over total cell area) is calculated and compared between TCP-transfected cells and *sfGFP* control. These data show that both MDA-MB-231 and Hs578t breast cancer cells that express either TCP-*VASH1* or TCP-*VASH2* degraded significantly more gelatin compared to those cells transfected with a control *sfGFP* vector (Figure 6B,D).

## 4. Discussion

The establishment of *VASH1* and *VASH2* coupled with *SVBP* to be the major TCPs has provided the opportunity to better understand the function of deTyr-Tub in various cell types. The necessity of both the vasohibin and *SVBP* to effectively remove the C-terminal tyrosine from α-tubulin likely contributed to the delay in the discovery of the TCP. Leveraging this discovery, we aimed to understand for the first time the effect of specifically enhancing deTyr-Tub in breast epithelial cells. Through this study, it was shown that addition of the TCP-containing constructs to both breast epithelial and tumor cell lines resulted in a dramatic increase in levels of filamentous deTyr-Tub (Figure 1). The conversion of Tyr-Tub to deTyr-Tub following transfection with the TCP constructs can be observed through immunofluorescence staining in Appendix A. By Western blot, the conversion of Tyr-Tub to deTyr-Tub can be observed, but the decrease is minimal given the heterogeneity of the transfection (Appendix A). Importantly, the increases in deTyr-Tub occur to the greatest extent with constructs expressing the vasohibin (*VASH1*/*2*) and its binding partner (*SVBP*), while expression of the vasohibin alone provides a smaller increase in deTyr-Tub (Figure 2). This increase in deTyr-Tub with vasohibin alone is likely occurring through the interaction with endogenous *SVBP* already expressed by the cells. Expression of *SVBP* alone does not lead to an increase in deTyr-Tub, indicating that the level of SVBP is not a limiting factor.

Our studies revealed that nontumorigenic breast epithelial cells and the modestly tumorigenic MCF7 cells did not tolerate the increase in deTyr-Tub and resulted in morphological changes consistent with increased apoptosis (Figure 3 and Figure 4). Given the knowledge that deTyr-Tub is increased at the invasive margin of breast epithelial carcinomas and elevated deTyr-Tub is associated with poor prognosis when combined with tumor grade, we sought to identify a molecular basis for tolerating enhanced deTyr-Tub through the induction of specific mutations in the background of breast epithelial cells.

To determine mechanisms of resistance, we first began by confirming that TCP-induced cell death was occurring through Caspase-3 activation in the immortalized nontumorigenic breast epithelial cell line MCF-10A (Figure 4). With this Caspase-3 activation confirmed following TCP overexpression, we utilized the High Content Analysis (HCA) feature of our Nikon Ti2e to identify GFP(+) cells, indicating a successful transfection of the constructs of interest. Using these identified cells, the HCA software then quantified the intensity of active Caspase-3, as labeled with a AF594 secondary antibody. The quantification of active Caspase-3 intensity was used to determine the percentage of transfected cells undergoing apoptosis across the different constructs and cell lines. 

Previously, our lab has generated a number of specific genetic changes in the stable background of the MCF10A cell line [33,34,35]. These altered cell lines allow us to determine how individual genetic alterations contribute to cancer phenotypes such as migration, dormancy, and metastasis [33,34,35]. Several of these changes are anti-apoptotic in function such as *PTEN*−/−, Bcl-2 and Bcl-xL overexpression, while others such as the activating KRas mutation alter migration and in vivo tumor formation. While exploring the effects of these mutations on deTyr-Tub, we discovered that the MCF10A KRas mutation has a modest effect on increasing this tubulin PTM alone or in combination with *PTEN*−/−. When transfecting the TCP constructs into the altered MCF10A cells, we anticipated that anti-apoptotic genes would provide the maximum protective effect against TCP-induced apoptosis. Previously, it has been reported that MCF10As with *PTEN* loss and Bcl-2 overexpression were more resistant to intrinsic apoptosis, as determined by placing cells in suspension, treatment with latrunculin A to induce cell rounding, and deprivation of essential growth factors [35]. Unexpectedly, *PTEN*−/− provided no protection against TCP-induced apoptosis and Bcl-2 and Bcl-xL overexpression cells only provided moderate protection (Figure 4). Meanwhile, KRas and *PTEN*−/− KRas cells were highly protected from TCP-induced apoptosis. Since the KRas and *PTEN*−/− KRas cells were developed from independent parental clones, this reinforces the conclusion that the protective effect from apoptosis induced by deTyr-Tub results from KRas activation, rather than clonal variation. While we cannot completely rule out that this apoptotic effect may be through the action of the TCP on another cellular process, we have increased confidence that it is due to its action on microtubules given the occurrence in both VASH1 and VASH2 TCP constructs. Of importance, VASH1 and VASH2 share only ~52.5% homology and have been shown to display differential functions in their roles with angiogenesis [39]. Therefore, we have increased confidence that their common function of generating deTyr-Tub is the cause of apoptotic induction.

When comparing components of the TCP across these cell lines, we discovered that *VASH1*, *VASH2*, or *SVBP* do not vary in the MCF10As and variants despite moderate changes in basal deTyr-Tub presence (Figure 5A). When comparing the abundance of deTyr-Tub across commonly utilized breast cancer cell lines, the level of the TCP components, *VASH1*, *VASH2*, and *SVBP*, did not show a clear correlation with the amount of deTyr-Tub (Figure 5B). Given the ability for the *VASH2* antibody to detect both *VASH1* and *VASH2* (Appendix A), we have increased confidence that the relative amount of vasohibin does not correspond to level of deTyr-Tub. The full, uncropped images for all Western blots can be found in Appendix A. This led us to conclude that the generation of deTyr-Tub may be controlled at an enzymatic regulation step rather than at the level of TCP protein abundance.

Prior research in our laboratory has revealed an increase in deTyr-Tub, “glu-tubulin” at that time, which occurs in combination with the EMT transcription factor, Twist, at the invasive margin of patient ductal carcinoma in situ samples [25]. These data suggested a functional relationship between deTyr-Tub and invasion. Additionally, we have observed that treating many different cell types with the microtubule stabilizing chemotherapeutic paclitaxel leads to a dramatic increase in the abundance of detyrosinated microtubules [40,41]. To better understand the connection between the enhanced deTyr-Tub observed at the invasive margin of patient samples, we began looking in the literature for a connection between established methods of increasing deTyr-Tub and mechanisms of invasion. A study by the Courtneidge group determined the effects of a wide variety of compounds on invadopodia formation [42]. Invadopodia are thought to be some of the earliest structures necessary for cells to escape the confines of the basement membrane [43]. This step is a requirement for breast cancers to be categorized as invasive [43]. The Courtneidge study revealed that breast cancer cells treated with paclitaxel lead to increased invadopodia formation [42]. Previously, we have observed that paclitaxel treatment greatly elevates the amount of deTyr-Tub [40,41], and therefore we sought to determine if directly increasing deTyr-Tub through the addition of TCP constructs will alter the invasive capability of breast cancer cells.

MDA-MB-231 cells harbor a KRas mutation as do the genetically engineered MCF10A KRas cells and both cell lines tolerate the transfection with TCP constructs. The Hs578t cell line contains an HRas mutation which may contribute to these cells tolerating elevated levels of deTyr-Tub. Interestingly, both the MDA-MB-231 and Hs578t cell lines are classified as Basal-B triple-negative breast cancer (TNBC) [44]. While there are many differences between the MCF10A KRas, MDA-MB-231, and Hs578t cell lines the activation of the Ras-pathway and survival following TCP transfection appear to be connected in a manner currently unknown to us. To better understand the impact of increasing deTyr-Tub in these cell lines, transfected cells were imaged every 15 min for 19 h, starting at 24 h post-transfection. The mean squared displacement (MSD) and mean speed were quantified for both MDA-MB-231s and Hs578t (Appendix A). The TCP constructs had differential effects on MSD and speed based on the cell line. These data suggest that deTyr-Tub may have an impact on migration, but more research must be done to understand its role and effect in this phenotype.

When plated on a gelatin matrix, MDA-MB-231 and Hs578t cells transfected with TCP constructs degraded the gelatin more effectively than cells transfected with the *sfGFP* control vector (Figure 6). The pattern of matrix degradation is consistent with the focal, punctate degradation that characterizes invadopodia, but we cannot conclusively determine that the increase in gelatin degradation is solely due to an increase in invadopodia. Cells at the end of the gelatin degradation assay were stained with phalloidin to look for co-localization of f-actin foci and gelatin voids (Appendix A). It is not yet known how deTyr-Tub increases gelatin degradation, but it is possible that deTyr-Tub could increase outward cell extension [45] or promote the recruitment of components to interface between the cell and extracellular matrix and elongation of invadopodia [46,47]. Recently, invadopodia have been shown to be a component in early cellular invasion and metastasis [48,49]. The interaction between microtubules, invadopodia, and invasion has been examined with findings suggesting that specific trafficking along microtubules may aid in the degradation of the extracellular matrix [50,51]. Alternatively, enhanced deTyr-Tub may stimulate programmed cellular mechanisms to enhance invasive capacity, or microtubules with enhanced deTyr-Tub may increase the outward force at established points of cell attachment. It is also possible that the increase in TCP activity in transfected cells may be altering their regulation of proteases necessary to degrade gelatin. Utilizing the published structure of the components of the TCP, especially the vasohibins harboring the site of enzymatic activity, there is currently great interest in developing new compounds to reduce the formation of deTyr-Tub. The evidence presented here indicates that TCP could be a potential therapeutic target to reduce metastatic phenotypes.

## 5. Conclusions

Our results reveal that when components of the TCP, *VASH1*/*2* plus *SVBP*, are expressed in breast epithelial cells, there is an increase in deTyr-Tub. Interestingly, the increase in deTyr-Tub does not persist as these nontumorigenic breast epithelial cells undergo apoptosis. The addition of oncogenic KRas, and to a lesser extent, Bcl-2 and Bcl-xL overexpression prevent these cells from undergoing apoptosis, allowing them to tolerate elevated levels of deTyr-Tub. Translating this discovery into the triple-negative breast cancer cell lines, MDA-MB-231 and Hs578t reveals that transfection of the TCP leads to increased focal degradation of the gelatin substrate.

These findings contribute to our understanding of the tubulin tyrosination/detyrosination cycle in breast epithelial cells, providing a new molecular target for microtubule-directed drugs. Current anti-microtubule chemotherapies globally stabilize (ex. taxanes) or destroy (ex. vinca alkaloids) cellular microtubules, and this broad disruption can reduce tumor growth but also yields substantial toxic side effects, including immune suppression and neuropathy. This study reveals that deTyr-Tub is tightly regulated in normal breast epithelial cells, and excess deTyr-Tub promotes apoptosis. These findings begin to reveal therapeutic opportunities to avoid the side effects of broad microtubule disruption by more precisely targeting the detyrosinated microtubule subset via TCP inhibition to reduce tumor progression and metastasis. Our current data support a model that specific targeting of TCP activity could help reduce the formation of invadopodia, especially in breast tumor cells where Ras activation promotes tolerance for high levels of deTyr-Tub.

## Figures and Tables

**Figure 1 cancers-14-01707-f001:**
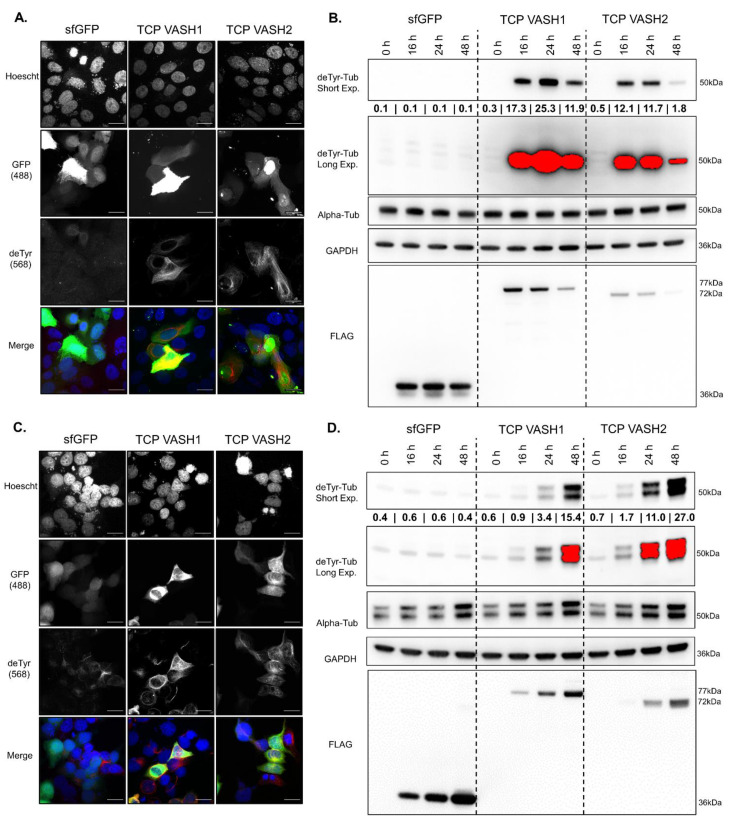
*VASH1*/*SVBP* and *VASH2*/*SVBP* increase deTyr-Tub in a specific and time-dependent manner. (**A**) MCF10A cells transfected with a plasmid containing *sfGFP*, *VASH1*-*sfGFP*-IRES-*SVBP*, or *VASH2*-*sfGFP*-IRES-*SVBP* and immunostained for deTyr-Tub (red) as well as the nuclear stain Hoescht 33342 (blue). Scale bar = 20 µm. (**B**) MCF10As were transfected with the same constructs as in panel A and lysed at times 0, 16 h, 24 h, and 48 h post-transfection. Primary antibodies against deTyr-Tub, α-tubulin, GAPDH, and FLAG-tag were incubated with HRP-conjugated secondary antibodies. *sfGFP*, *VASH1*, and *VASH2* expressed on the plasmids are FLAG-tagged, allowing for the simultaneous visualization of all three proteins at various molecular weights. (**C**) MCF7 cells were transfected with the same plasmids as in panel A and processed in the same manner for visualization of deTyr-Tub-rich microtubules. (**D**) MCF7 cells were transfected with the same constructs as in panel A, and probed for the protein targets as in panel B. The values under deTyr-Tub indicate the raw intensity/1000 for the band above without any normalization.

**Figure 2 cancers-14-01707-f002:**
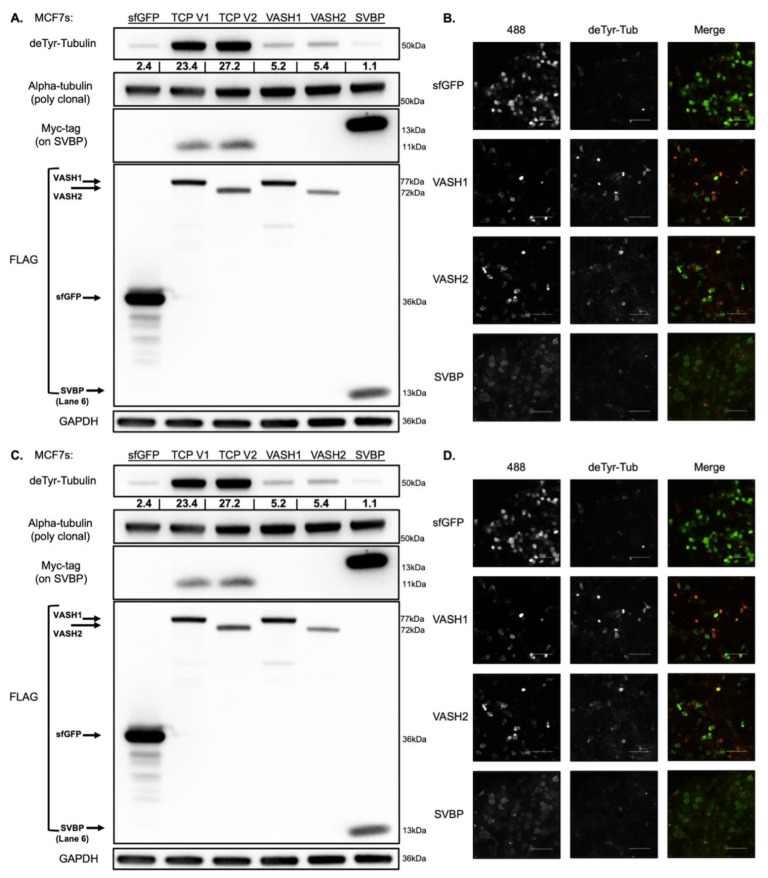
TCP-*VASH1* and TCP-*VASH2* plasmids lead to the maximal deTyr-Tub increase in MCF10A human mammary epithelial cells and MCF7 breast cancer cells. (**A**) MCF10A cells were plated and transfected with a CAG plasmid containing TCP-*VASH1*, TCP-*VASH2*, *VASH1*-*sfGFP* alone, *VASH2*-*sfGFP* alone, or *SVBP* alone for 24 h prior to being lysed with RIPA buffer. The TCP-*VASH1* or TCP-*VASH2* constructs generate the highest level of deTyr-Tub after 24 h. SVBP did not increase deTyr-Tub compared to *sfGFP* control, while *VASH1* and *VASH2* alone generated an intermediate increase in deTyr-Tub. Probing with the FLAG-tag antibody reveals the presence of *sfGFP*, vasohibin, or *SVBP* alone. The expression of *SVBP* can be visualized in both the *SVBP* alone and IRES-expressed TCP constructs given the myc-tag on *SVBP*. (**B**) Immunofluorescence staining for deTyr-Tub revealed that the increase in deTyr-Tub with vasohibin alone is filamentous, but not present in all transfected cells. SVBP transfection was determined by using a primary antibody (1:1000) to stain for the c-Myc tag on *SVBP* and a secondary 488 antibody (1:1000) to determine positive cells. No change in deTyr-Tub is observed in the cells transfected with *SVBP* alone. (**C**) MCF7 cells were plated and transfected with the same plasmid constructs as in panel A with collection of the protein lysate 24 h after transfection. The lysates were run on a Western blot and probed for the same protein targets as in panel A. (**D**) Immunofluorescence imaging of MCF7 cells transfected with the individual components of the TCP, and stained for deTyr-Tub filaments in the same manner as panel B. The values under deTyr-Tub indicate the raw intensity/1000 for the band above without any normalization.

**Figure 3 cancers-14-01707-f003:**
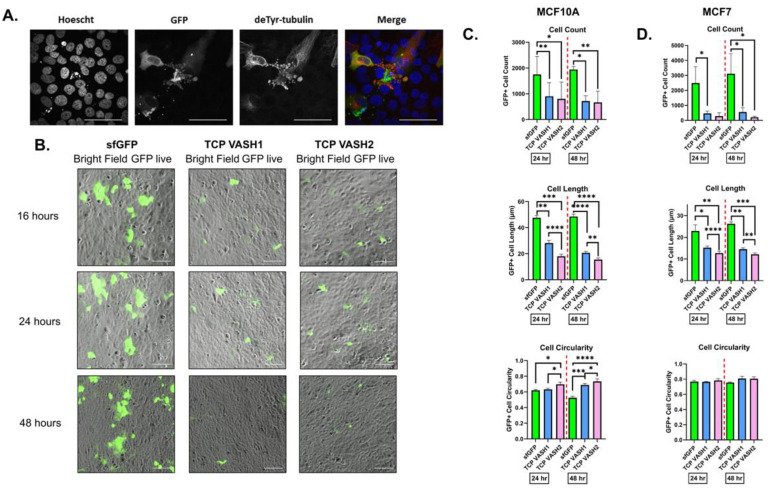
TCP expression induced cell death in MCF10s and MCF7s. (**A**) MCF10A cells transfected with a TCP-*VASH1* construct was stained for deTyr-Tub (red, 1:1000) and the nuclei stained with Hoechst 33342 (blue, 1:5000). This image shows nuclear condensation and the formation of apoptotic-like bodies which are positive for the *VASH1*-sfGFP fusion protein as well as deTyr-Tub. Scale bar = 50 µm (**B**) MCF10A cells were imaged 16, 24, and 48 h post-transfection in both phase contrast and the GFP channel. Representative images of the GFP channel overlaid on phase contrast show cell rounding in the TCP-*VASH1* and TCP-*VASH2*, but not in the *sfGFP* control. Scale bar = 100 µm (**C**) Using the HCA Nikon Analysis software, *sfGFP*-positive objects were identified. To compare condition, the number of objects, length and circularity of each object were quantified. TCP-transfected MCF10A cells showed a lower cell number, decrease in cell size, and an increase in circularity compared to the *sfGFP* control. (**D**) MCF7 cells showed a decrease in cell number, and size for TCP transfections compared to *sfGFP* control. There was no significant difference in cell circularity amount constructs in MCF7 cells. * *p* < 0.05, ** *p* < 0.01, *** *p* < 0.001, and **** *p* < 0.0001, *n* = 4; triplicate.

**Figure 4 cancers-14-01707-f004:**
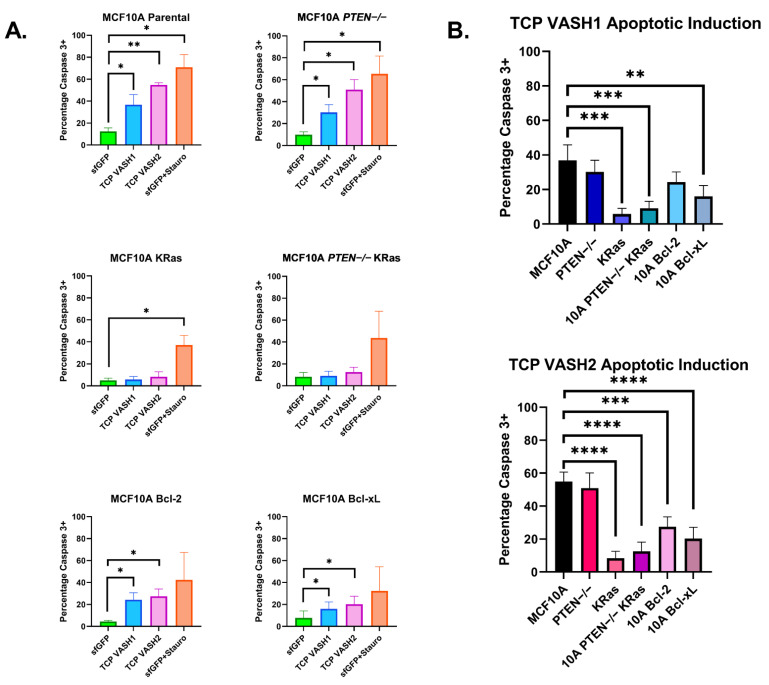
TCP transfection induces apoptosis that is blocked by oncogenic KRas, Bcl-2 and Bcl-xL overexpression. (**A**) Specific oncogenic or anti-apoptotic signaling pathways were manipulated in the stable background of MCF10A cells. These cell lines were then transfected with sfGFP control, TCP-*VASH1*, or TCP-*VASH2* for 24 h. After 24 h, the cells were fixed and stained with an anti-active Caspase-3 antibody and AF594-conjugated secondary to determine cells actively undergoing apoptosis. Using the HCA Nikon Analysis software, *sfGFP*-positive cells were identified and then average intensity of the AF549 signal for each *sfGFP*-positive object was determined. If the AF549 average intensity was greater than 10,000 RFU, that object was counted as positive for active Caspase-3. Cells transfected with TCP-*VASH1* and TCP-*VASH2* had a higher level of Caspase-3 activation compared to the sfGFP control in MCF10A parentals, *PTEN*−/−, Bcl-2- and Bcl-xL-overexpressing cells. Interestingly, MCF10A KRas and *PTEN*-/ KRas did not show an increase in Caspase-3 activation compared to *sfGFP* control. (**B**) For each TCP construct (*VASH1* or *VASH2*), the level of Caspase-3 activation was compared to *sfGFP*. Oncogenic *KRas* provides the most protection against TCP-induced cell death, but Bcl-2 and Bcl-xL also provide a lesser level of protection. * *p* < 0.05, ** *p* < 0.01, *** *p* < 0.001, and **** *p* < 0.0001, *n* = 4; triplicate.

**Figure 5 cancers-14-01707-f005:**
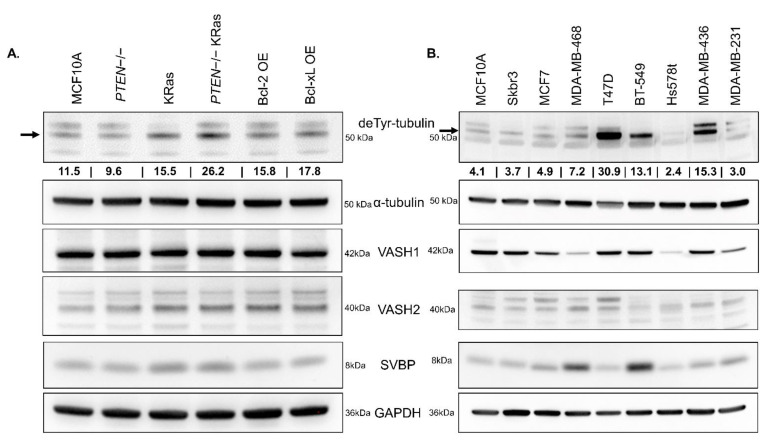
Components of the TCP across cell lines do not correlate with the level of deTyr-Tub. (**A**) Protein was isolated from MCF10As and the MCF10A variants using RIPA lysis buffer. DeTyr-Tub (1:10,000), total alpha-tubulin (1:1000), and the components of the TCP (*VASH1*(1:1000), *VASH2* (1:1000), and *SVBP* (1:1000)) were probed. An increase in basal level of deTyr-Tub can be seen in the MCF10A cell lines with oncogenic *KRas*, that is not present in the MCF10A parental or anti-apoptotic MCF10A cell lines. The level of *VASH1* and *VASH2* appears to be consistent across all lines, with a small increase in *SVBP* in the *KRas* containing lines. (**B**) To better understand the presence of deTyr-Tub and TCP components in breast cancer, a number of commonly used breast cancer cell lines were lysed using RIPA buffer and subjected to immunoblotting. These breast cancer cell lines contain various basal amounts of deTyr-Tub, *VASH1*, *VASH2*, and *SVBP* with no clear connection between components of the TCP and level of deTyr-Tub. The values under deTyr-Tub indicate the raw intensity/1000 for the band above without any normalization.

**Figure 6 cancers-14-01707-f006:**
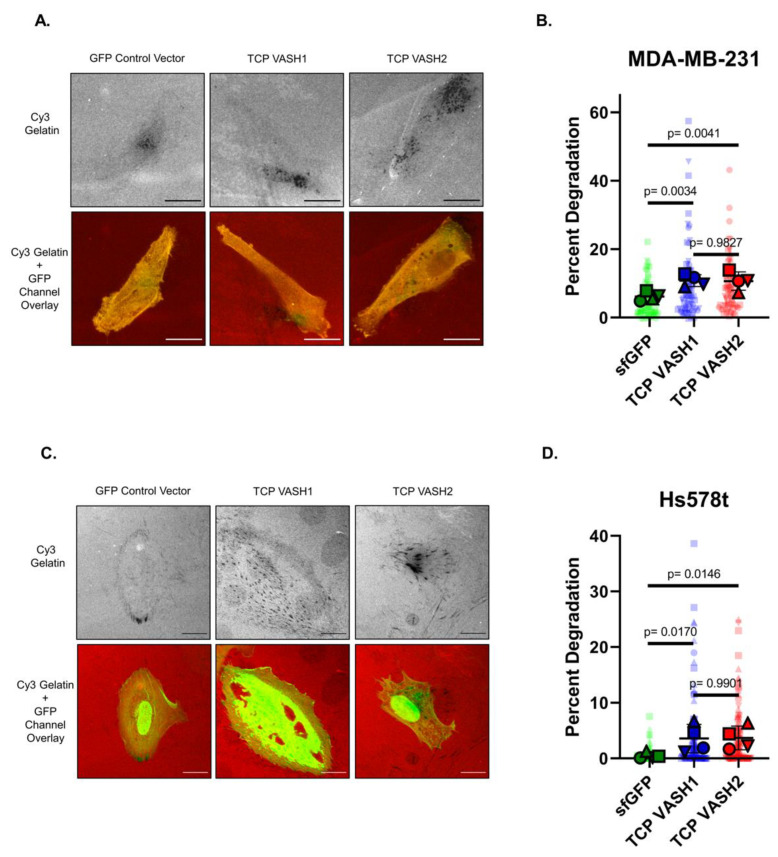
Increased deTyr-Tub via TCP expression in MDA-MB-231 and Hs578t cells leads to an increase in gelatin degradation. (**A**) MDA-MB-231 breast cancer cells were transfected with constructs containing sfGFP control, TCP-*VASH1*, or TCP-*VASH2* for 24 h prior to being plated on Cy3 gelatin for another 24 h. Cells were fixed and imaged on an Olympus FV1000 confocal microscope. Representative images are shown in the upper row as a gray scale image of the gelatin, with dark areas indicating the area of gelatin degradation for sfGFP control, TCP-*VASH1*, or TCP-*VASH2*. The lower row of images shows the cell body in green and gelatin in red. Scale bar = 20 µM. (**B**) The percentage of gelatin degraded per cell area was determined from 10 cells per in duplicate or triplicate from 4 biological replicates. Each light-colored symbol corresponds to a single image, outlined shapes are the mean for each biological replicate, and the horizonal bar is the mean for all values. Cells transfected with TCP-*VASH1* or TCP-*VASH2* degrade significantly more gelatin than those transfected with *sfGFP*. (**C**) Hs578t breast cancer cells were transfected in the same manner as in panel A, but allowed to degrade the gelatin for 48 h and representative images are shown. Scale bar = 20 µM. (**D**) Images of Hs578t cells were processed and graphed in the same manner as panel B. Hs578t cells transfected with TCP-*VASH1* or TCP-*VASH2* degrade significantly more gelatin than those transfected with *sfGFP*.

## Data Availability

The data will be freely provided upon request from corresponding author.

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
