# Peer review of "Tubulin Carboxypeptidase Activity Promotes Focal Gelatin Degradation in Breast Tumor Cells and Induces Apoptosis in Breast Epithelial Cells That Is Overcome by Oncogenic Signaling"

_cancers, 2022, doi:10.3390/cancers14071707_

Round 1

Reviewer 1 Report

The authors have substantially improved the quality of their work. The article is ready to be accepted for publication in its present form.

Reviewer 2 Report

Modifications made should be enough for publishing.

PD. Potential typo in line 124

This manuscript is a resubmission of an earlier submission. The following is a list of the peer review reports and author responses from that submission.

Round 1

Reviewer 1 Report

In this article,  Mathias et al. evaluate the effect of TCP overexpression in breast cancer cells and breast epithelial cells. Although of interest, some major and minor issues should be addressed prior to final acceptance of the paper.

MAJOR ISSUES

1) Levels of Tyr-Tubulin should me also measured in both Western-blot and immunofluorescence studies.

2) The authors claim that TCP gain-of-function results in an increase of invadopodia formation. However, an increment in the degradation of gelatin itself is not enough to fully demonstrate an increase in the formation of invadopodia. It could be argued that TCP overexpression results in an increase in the expression of gelatinases such as MMP2 or MMP9. More experiments are needed to make this statement.

3) It would be very helpful to evaluate the effect of the increase of TCP activity in the migratory capacity of the cells.

4) The authors must demonstrate the oncogenic expression of KRAS G12V in the cells by pull down assays or GLISA assays. In this regard, it is not clear what is the physiological relevance of oncogenic KRASG12V expression to rescue the apoptotic phenotype. Are the authors suggesting that TCP overexpression results in a reduction of RAS-mediated signaling pathway? Moreover, this mutation is very rare in breast cancer. For example, although during the discussion the authors suggest that MDA-MB-231 harbor a “similar” mutation that KRAS, that is not true. These cells hold high levels of mutations in p53. In this context, have the authors examine whether TCP overexpression in these cells (without oncogenic KRAS) induce cell death?

5) To fully demonstrate that the phenotype observed in the cells upon TCP transfection is directly linked to the increase of deTyr-Tub levels, it would be necessary to revert the phenotype by increasing the activity of TTL in these cells.

MINOR ISSUES

1) Other works have named/identified other enzyme known as NNA1/AGTPBP1/CCP1 that act as TubCP. Is the expression of this protein altered by the overexpression of VASH proteins? Could the authors clarify to the reader the differences between VASH and NNA1/CCP1 through the Introduction section?

Reviewer 2 Report

Mathias et al. studied the role of TCP-induced-tubulin de-tyronisation(De-Tyr-Tub) in breast cancer. They describe a pro-apoptotic role of De-Tyr-Tub found in MCF10A that is attenuated upon certain oncogenic stimuli, which explains the lower sensitivity of cancer cells to this modification. Moreover, they show a correlation between de-Tyr-Tub and cell invasion trough gelatin degradation assays, suggesting a potential link with a metastatic behavior in vivo.

The paper is coherent, well-explained and claims are well-documented. However, some doubts arise after reading it:

Main points

Can authors clearly state that De-Tyr-Tub is the responsible for cell death and no other potential effects of TCP. In other words, how do authors know that TCP exclusively act on Tubulin?

How authors explain that TCP expression reduced cell count in MCF7 even more than in MCF10A? (Fig. 3C and 3D). Are they proliferating less but dying less? Can authors provide a quantitative measurement of these parameters upon TCP expression in MCF10A and MCF7. It would actually be interesting to extend this to MCF10A with the different oncogenes, especially KRas.

In Figure 6 differences are very slight. Have authors thought about other assays such as migration or invasion assays in transwells for example? This may allow the study of the invasion capacity in parental and KRAS-expressing MCF10A or to compare the effect of de-Tyr-Tub in high and low TCP expression cell lines (anticipating that TCP expression would have a more drastic effect on cells with a low de-Tyr-Tub)  

Minor points

Figures are not always mentioned in order (meaning A-B-C-D), such in Figure 1, where A and C are called before B and D.  

In line 476 the statement “However, TCP-VASH1 expression in the cells expressing the oncogenic KRas (alone or in combination with PTEN loss)and overexpressing Bcl-xL, but not Bcl-2, showed a significant decrease in the amount of Caspase-3 positivity compared to the MCF10A cells.” Is miss leading. There is a lack of statistical significance, but data from Bcl-xL and Bcl2 upon TCP VASH1 expression are fairly similar and actually.

Can authors provide quantification for de-Tyr-Tub in Figure 5? The increase is not well appreciated, at least in that blot.

Author comment on the potential post-translational regulation of TCP. Have they tried to correlate de-Tyr-Tub with the phosphorylation, acetylation… of the proteins or even the amount of complex in the cell??